# Melatonin Ameliorates Cadmium Toxicity in Tobacco Seedlings by Depriving Its Bioaccumulation, Enhancing Photosynthetic Activity and Antioxidant Gene Expression

**DOI:** 10.3390/plants13213049

**Published:** 2024-10-31

**Authors:** Abdul Ghaffar Shar, Sadam Hussain, Muhammad Bilawal Junaid, Maqsood Ul Hussan, Usman Zulfiqar, Amal Mohamed AlGarawi, Rafal Popielec, Lixin Zhang, Arkadiusz Artyszak

**Affiliations:** 1College of Life Sciences, Northwest A&F University, Yangling 712100, China; abdulghaffarshar@hotmail.com; 2College of Horticulture, Northwest A&F University, Yangling 712100, China; ch.sadam423@gmail.com; 3Department of Plant Production, College of Food and Agriculture, King Saud University, P.O. Box 2455, Riyadh 11451, Saudi Arabia; 4College of Grassland Agriculture, Northwest A&F University, Yangling 712100, China; maqsoodg73@gmail.com; 5Department of Agronomy, Faculty of Agriculture and Environment, The Islamia University of Bahawalpur, Bahawalpur 63100, Pakistan; usmanzulfiqar2664@gmail.com; 6Department of Botany and Microbiology, College of Science, King Saud University, P.O. Box 2455, Riyadh 11451, Saudi Arabia; aalgarawi@ksu.edu.sa; 7Institute of Agriculture, Warsaw University of Life Sciences-SGGW, Nowoursynowska 159, 02-776 Warsaw, Poland

**Keywords:** antioxidants, gas exchange attributes, melatonin, remediation, heavy metals, qRT-PCR, florescence, Cd accumulation

## Abstract

Soil remediation for cadmium (Cd) toxicity is essential for successful tobacco cultivation and production. Melatonin application can relieve heavy metal stress and promote plant growth; however, it remains somewhat unclear whether melatonin supplementation can remediate the effects of Cd toxicity on the growth and development of tobacco seedlings. Herein, we evaluated the effect of soil-applied melatonin on Cd accumulation in tobacco seedlings, as well as the responses in growth, physiological and biochemical parameters, and the expression of stress-responsive genes. Our results demonstrate that melatonin application mitigated Cd stress in tobacco, and thus promoted plant growth. It increased root fresh weight, dry weight, shoot fresh weight and dry weight by 58.40%, 163.80%, 34.70% and 84.09%, respectively, compared to the control. Physiological analyses also showed significant differences in photosynthetic rate and pigment formation among the treatments, with the highest improvements recorded for melatonin application. In addition, melatonin application alleviated Cd-induced oxidative damage by reducing MDA content and enhancing the activities of enzymatic antioxidants (CAT, SOD, POD and APX) as well as non-enzymatic antioxidants (GSH and AsA). Moreover, confocal microscopic imaging confirmed the effectiveness of melatonin application in sustaining cell integrity under Cd stress. Scanning Electron Microscopy (SEM) observations illustrated the alleviative role of melatonin on stomata and ultrastructural features under Cd toxicity. The qRT-PCR analysis revealed that melatonin application upregulated the expression of photosynthetic and antioxidant-related genes, including *SNtChl*, *q-NtCSD1*, *NtPsy2* and *QntFSD1*, in tobacco leaves. Together, our results suggest that soil-applied melatonin can promote tobacco tolerance to Cd stress by modulating morpho-physiological and biochemical changes, as well as the expression of relevant genes.

## 1. Introduction

In recent years, heavy metal contamination has become increasingly severe due to rapid industrialization and the land application of sludge waste [1]. Heavy metal-contaminated sites are harmful not only to soil biota but also to plant cultivation and production, ultimately threatening ecological and food security [2]. High concentrations of heavy metals in soil inhibit seed emergence, crop growth and development, and reduce crop yield by disrupting plant cellular functions and inducing oxidative damage in plants [3]. Among the major toxic and mobile elements, cadmium (Cd) is recognized as one of the most serious contaminants impacting agricultural lands and the environment [4]. According to an estimate, soil metal pollution sites are increasing at a rate of 16%, with Cd toxicity contributing significantly to this rise [5]. In China, Cd toxicity in cropland soils has become a major obstacle to achieving green agricultural goals [6]. Thus, managing Cd-contaminated sites is crucial for promoting sustainable agricultural development.

Tobacco, an important member of the Solanaceae family, is cultivated as an annual herbaceous plant [7]. It is a significant model and economic crop worldwide, including in China, where it is cultivated in almost all provinces [8]. In China, approximately 1.01 million hectares are dedicated to tobacco cultivation, producing around 2.13 million tons each year [7]. Tobacco plays a significant role in the remediation of heavy metal-polluted sites due to its high capability for metal uptake compared to other crops [9]. However, previous studies have shown that tobacco is particularly susceptible to heavy metals accumulation, especially Cd [10,11]. The elevated uptake of the metals poses a significant risk to human health due to the inhalation of cigarette smoke. Moreover, excessive Cd uptake can impair the plant’s metabolic functions, leading to reduced chlorophyll synthesis and an overall decline in photosynthetic efficiency [12]. These physiological impacts are evident in stunted seedling growth and decreased nutrient uptake under Cd toxicity. In addition, high Cd levels in the rhizosphere and accumulation in plant tissues can disrupt the antioxidative defense system, facilitating ROS-induced oxidative damage [8,10]. Collectively, these effects lead to a significant reduction in tobacco agronomic yield and economic returns. Therefore, remediating Cd-contaminated sites is critical to ensuring sustainable tobacco production and improving overall soil and environmental health [12].

In recent years, various agronomic measures, including nutrient management through foliar sprays of micronutrients, controlled irrigation practices, and the cultivation of high-tolerant crop cultivars, have been adopted to address Cd toxicity in cropland soils [13,14,15]. Melatonin, known as a stress-mitigating agent, is gaining attention for its ability to enhance crop performance under heavy metal stress conditions [16]. As a natural compound, melatonin has the potential to stimulate crop growth and development even in the presence of biotic and abiotic stresses [17]. The improvement in plant growth observed with melatonin application can be attributed to enhanced leaf photosynthesis, the higher accumulation of osmoprotectants, and improved leaf senescence [18,19]. Similarly, enhanced seedling growth has been positively correlated with improved leaf photosynthesis, as observed in pepper seedlings exposed to chilling stress [20]. Moreover, the foliar application of melatonin has been shown to substantially increase osmolyte accumulation, thereby enhancing overall crop performance under low light conditions [21]. Improved crop growth under melatonin treatment may also be associated with higher nutrient accumulation. Melatonin also interacts with other plant hormones, balancing growth and stress responses to support plant resilience in adverse environments. It has been shown that melatonin influences lignin deposition in plants and enhances their structural integrity. Lignin, as a major constituent of the plant cell wall, offers mechanical strength and protection against stress stimuli. Furthermore, melatonin application promotes the activities of enzymes crucial for mitigating the harmful effects of ROS and oxidative stress during stressful conditions [16]. Improved physiological responses, along with increased enzymatic and non-enzymatic activities, have been well documented under heat stress [22], chilling [20], arsenic [23], and nutrient toxicity [24]. However, while the mitigating effects of melatonin on abiotic stresses have been reported [25], the remediation potential of melatonin application on Cd-contaminated sites requires more in-depth studies, particularly focusing on the physio-biochemical and molecular aspects of the tobacco crop [26].

This study explored the influence of soil-applied melatonin on tobacco growth, Cd uptake and translocation, photosynthetic and biochemical aspects, and related gene expression. Our major objectives were: (1) to assess the growth responses of tobacco plants under Cd toxicity following melatonin application; (2) to elucidate the physiological and biochemical mechanisms by which soil-applied melatonin enhances tobacco resistance to Cd toxicity; and (3) to determine the relative expression of genes related to photosynthesis and enzymatic activities in tobacco exposed to Cd stress and melatonin treatment. We hypothesized that soil-applied melatonin would mitigate Cd toxicity by limiting Cd uptake and translocation in tobacco tissues, promoting seedling growth, and regulating physio-biochemical indices.

## 2. Results

### 2.1. Growth Indices

Cd and melatonin application significantly affected the growth of tobacco plants (Figure 1 and Appendix A). Tobacco seedlings treated solely with Cd exhibited a decline in growth indices, including seedling fresh and dry weight, compared to other treatments. Cd alone reduced root fresh weight, root dry weight, shoot fresh weight and shoot dry weight by 32.39%, 38.59%, 22.95%, and 37.13%, respectively, lower than the control plants. On the other hand, melatonin supplementation significantly improved these traits under both control and Cd stress conditions. Specifically, melatonin application under Cd stress improved root fresh weight, root dry weight, shoot fresh weight and shoot dry weight by 58.40%, 163.80%, 34.70% and 84.09%, respectively, compared to the control treatment without melatonin and Cd application. Overall, in terms of increasing growth traits, the treatments were ordered as melatonin > melatonin + Cd > control > Cd.

### 2.2. Photosynthetic Indices

There was a significant difference in photosynthetic indices, except Fv/Fm, among the melatonin and Cd treatments (Figure 2). The Pn, Gs, Tr, Ci, and chlorophyll contents of seedlings exposed only to Cd were significantly lower than those in the control and melatonin-treated groups. Compared to control, Cd treatment decreased Pn, Gs, Tr, Ci, and chlorophyll content by 55.17%, 73.61%, 37.06%, 15.69%, and 38.11%, respectively. However, under Cd stress conditions, melatonin treatment significantly improved the values of these traits by 91.60%, 121.05%, 65.87%, 21.07%, and 58.08%, respectively, compared to the control group. These findings suggest that melatonin application enhances photosynthesis and pigment formation in tobacco under Cd stress conditions.

### 2.3. Cadmium Accumulation

Cd accumulation in roots and shoots as well as the bioconcentration factor (BCF) for both organs was significantly affected by Cd and melatonin treatments (Figure 3). Under Cd stress, melatonin application greatly reduced Cd accumulation in roots and shoots by 27.54% and 34.35%, respectively, lower than the sole Cd treatment. Similarly, tobacco seedlings treated solely with Cd showed an increase in BCF in roots and shoots. Nonetheless, melatonin application under Cd stress markedly decreased BCFs by 25.46% and 43.20% in roots and shoots, respectively, compared to the Cd only treatment.

### 2.4. Reactive Oxygen Species and Confocal Imaging

Compared with the control and melatonin groups, sole Cd supplementation substantially increased MDA, H_2_O_2_, O_2_^•−^ contents and EL in shoots (Figure 4). However, melatonin application significantly decreased these values, especially when applied alone or under Cd stress conditions. In Cd-treated plants, melatonin supplementation decreased MDA, H_2_O_2_ and O_2_^•−^ contents and EL by 35.87%, 25.79%, 26.91% and 30.93%, respectively, compared with the Cd-only group. The effectiveness of our treatments in reducing these indices ranked as follows: melatonin < melatonin + Cd < control < Cd.

In addition, confocal visualization for dihydroethidium (DHE) staining was carried out to assess ROS production in tobacco leaves (Figure 5). Confocal images showed intense fluorescence for the sole Cd treatment, indicating high accumulation of O_2_^•−^. On the other hand, melatonin application under Cd stress showed relatively reduced fluorescence, suggesting lower accumulation of O_2_^•−^ compared to the sole Cd treatment. These observations demonstrate the high efficacy of melatonin application in reducing ROS accumulation in tobacco leaves.

### 2.5. Enzymatic and Non-Enzymatic Antioxidants

Compared with the control, significantly higher activities of antioxidative enzymes, including CAT, POD, SOD and APX, were observed for the sole Cd group (Figure 6). However, melatonin application led to a significant further increase in the activities of these enzymes, boosting CAT, SOD, POD and APX activities by 57.30%, 15.26%, 15.67% and 116.78%, respectively, in Cd-treated seedlings compared to the control group. Analyses of non-enzymatic activities showed that sole Cd treatment significantly increased GSH and AsA values compared to the control group (Figure 7). However, melatonin supplementation further enhanced GSH and AsA activities, with marked improvements in the Cd-treated group. Under Cd stress, melatonin application increased GSH activity by 176.73% and AsA activity by 137.11% compared to the control treatment without melatonin and Cd.

### 2.6. SEM Analysis

Cd accumulation in above-ground plant parts can damage the leaf epidermis and stomata (Figure 8). SEM analysis was performed to analyze the influence of melatonin supplementation on guard cells and stomatal aperture. Recognizable effects were observed on the leaf epidermis, with melatonin application prominently affecting guard cell structure and stomatal aperture (Figure 8). Cd ion accumulation significantly reduced stomatal opening, with most stomata undergoing complete closure due to plasmolysis of the stomatal cells.

### 2.7. Response of Chlorophyll and Antioxidative Genes to Melatonin Application

There was a significant difference in the expression of pigment and antioxidative genes among the melatonin and Cd treatments. Both Cd and melatonin applications significantly influenced the expression of the SNtChl, q-NtCSD1, NtPsy2, and QntFSD1 genes in tobacco leaves (Figure 9). The highest expression levels of these genes were observed with melatonin application under Cd stress, followed by sole melatonin and Cd applications, compared to the control group. In Cd-treated plants, melatonin application enhanced the expressions of SNtChl, q-NtCSD1, NtPsy2, and QntFSD1 genes by 104.91%, 37.66%, 64%, and 91.07%, respectively, compared to the control group.

### 2.8. Mantel Analysis

A mantel correlation was performed to analyze relationships among the recorded parameters. As shown in Figure 10, seedling fresh and dry weights were positively correlated with photosynthesis, chlorophyll content and fluorescence. These growth indices also demonstrated a strong positive association with both enzymatic and non-enzymatic antioxidants. Leaf gas exchange indices, such as Pn, gs, Tr and Ci, were positively correlated with growth traits. Growth traits were strongly and positively associated with antioxidant enzyme activities but negatively associated with ROS and MDA content. Leaf gas exchange traits and chlorophyll fluorescence demonstrated a strong negative association with stress indicators.

## 3. Discussion

Remediation of Cd-contaminated sites, particularly through organic methods, is a hot topic of research because Cd poses detrimental effects on soil environments and agronomic crop yields [27,28]. As such, the application of phytohormones has gained considerable attention as a means to enhance remediation efforts against Cd pollution. Melatonin, as a growth stimulator, offers numerous benefits to plant growth and development under stress conditions [28]. It enhances seedling tolerance to Cd stress by improving growth, as well as physiological and biochemical indices. In this work, our results indicated that Cd stress significantly inhibited the growth of tobacco seedlings. However, the soil application of melatonin notably promoted seedling growth under Cd stress conditions. Similarly, numerous published reports demonstrated the negative consequences of Cd on the seedling growth of many species [29]. On the other hand, melatonin application has been widely shown to enhance plant growth under HM stresses. For example, Sun et al. [30] reported that melatonin application markedly enhanced sunflower growth under chromium toxicity, which might be associated with ionic homeostasis and better scavenging ability. In another study, Altaf et al. [31] reported the positive effects of melatonin on vanadium-exposed watermelon seedlings, where significant improvements in growth indices were observed. These improvements were associated with reduced accumulation of metal ions and the increased biosynthesis of melatonin under stress conditions. Similarly, Charoenphun et al. [20] demonstrated that melatonin improved the growth performance of pepper seedlings, potentially due to modulated N metabolism and the activity of chelating agents under arsenic toxicity. In line with our results, improved plant growth under melatonin application in Cd-stressed tomatoes was reported by Song et al. [32], with the improvement linked to enhanced physiological indices. Altogether, improved seedling growth under melatonin treatment can be explained in several ways: first, melatonin application effectively adjusts root configuration and improves ionic homeostasis under HM toxicity [16,28,33], enhancing nutrient absorption and utilization, which supports seedling growth through modified root structure. Second, melatonin facilitates the secretion of organic acid anions, reducing the uptake of metal ions [34]. Third, it improves nutrient accumulation in stressful environments [35].

The photosynthetic balance of plants is crucial for their survival, particularly under stress conditions. Cd stress has negative consequences on plant physiological indices, as reported in this study. Under Cd stress, the higher accumulation of Cd in plants tissues, particularly in leaves, leads to an inhibited supply of assimilates and impaired leaf functions. Our results demonstrate that sole Cd application drastically reduced chlorophyll pigment formation and leaf photosynthetic rates. According to Tan et al. [12], reduced photosynthesis under Cd stress is mainly linked to its damaging effects on light-harvesting complexes. On the other hand, Song et al. [32] reported that Cd disrupted chloroplast structure and reduced chlorophyll synthesis, which, in turn, led to decreased photosynthesis. Moreover, Cd negatively affected stomatal conductance and intercellular CO_2_ concentration, thereby inadequately fulfilling the prerequisites for photosynthesis [36]. Also, under Cd stress, a strong association between nonstomatal limitations and leaf photosynthetic rates has been well reported (Song et al. [32]; Figure 10). On the other hand, melatonin application under both control and Cd stress conditions noticeably improved physiological traits in tobacco. Charoenphunet al. [20] similarly reported active photosynthetic processes in pepper under melatonin treatment. According to Yang et al. [28], the improved photosynthetic rates in melatonin-treated leaves were linked to its role in maintaining the integrity of D1 protein, which is an essential component of photosystem II. Additionally, Sun et al. [37] reported that melatonin facilitates the accumulation of mineral ions, which enhances the function of antioxidative enzymes, supports photosynthesis, and delays leaf aging under stress conditions. According to Altaf et al. [38], melatonin supplementation maintained the activity of photosynthetic machinery, which was associated with reduced damage to the thylakoid membrane. Enhanced pigment formation and photosynthesis under melatonin application has been well documented in response to Cu stress by Rai et al. [39] and Cd stress by Kano et al. [40]. Herein, to gain further insights into the physiological mechanisms improved by melatonin, we assessed the expressions of several relevant genes, including SNtChl and NtPsy2. The soil application of melatonin significantly upregulated the expression of these genes in tobacco under Cd stress. Similarly, Altaf et al. [31] demonstrated that melatonin exposure led to the high expression of photosynthetic-related genes, such as CB12 and CAB7, in pepper seedlings. The active role of these genes in chlorophyll pigment formation has been well documented by Ahmad et al. [41].

Sole cadmium treatment also facilitated oxidative damage by causing an imbalance between ROS production and the activities of antioxidative enzymes (Figure 6). Under Cd toxicity, excessive accumulation of ROS in plant cells can agitate cell membrane integrity and disrupt its function. Moreover, excessive ROS production is a leading cause of lipid peroxidation and impaired pigments’ development [21]. However, melatonin supplementation is known to help maintain the balance between ROS production and enzymatic activities. In this work, melatonin application significantly improved both enzymatic and non-enzymatic activities, while it decreased ROS production and MDA content in tobacco seedlings exposed to Cd stress (Figure 4, Figure 5 and Figure 6). Similar results were reported by Yu et al. [42], who found that melatonin application significantly decreased ROS production in rice seedlings under salt stress. Melatonin effectively mitigates the toxic effects of cadmium [28]. Recently, Malik et al. [18] depicted that melatonin application improved antioxidant activities, thereby facilitating redox homeostasis and mitigating oxidative stress induced by chromium toxicity. These studies underscore the important role of melatonin in alleviating the adverse effects of heavy metals, including Cd stress, on the plant antioxidant system. In line with these findings, our study indicates that melatonin application improves Cd tolerance in tobacco by improving antioxidative activities and reducing ROS accumulation. This is further supported by gene expression data and confocal imaging showing higher ROS accumulation under Cd stress. Furthermore, published reports have shown that Cd stress can elevate GSH and AsA levels, which are critical for cellular redox buffering [43]. It has also been well documented that elevated GSG levels accelerate the synthesis of chelator proteins, which are essential for maintaining ROS balance in plant cells [44]. Herein, Cd stress increased GSH levels in tobacco leaves, and melatonin supplementation further increased GSH activity. Similar results were reported in previous studies, where Xu et al. [45] observed a significant increase in GSH activity in *Siegesbeckia orientalis* (L.) and Wang et al. [46] in *Malus domestica* under Cd toxicity with melatonin application. For these mechanisms, numerous genes play regulatory roles to enhance plant defense. In this study, melatonin application under Cd stress resulted in the expression of the Cu/Zn superoxide dismutase gene NtCSD1 and the superoxide dismutase genes *NtFSD1a/b/c*. These findings align with those of Fu et al. [47], who reported increased expressions of the *NtSOD*, *NtPOD*, and *NtAPX* genes under salt stress.

## 4. Materials and Methods

### 4.1. Experimentation and Treatments

In this work, the tobacco variety Qingxue 103, cured in our laboratory, was utilized. Initially, plants were grown in a growth room until reaching the three-leaf stage. Later, they were transferred to a greenhouse and grown in soil-filled pots, each containing 8 kg of soil. One healthy seedling was grown in each pot. The experimental soil (with the following properties: bulk density = 1.24 ± 0.2 g·c^−3^; SOM = 11.96 ± 0.89 g·kg^−1^; available N, P and K were 52.99 ± 2.38, 22.28 ± 1.31, and 96.98 ± 4.60 mg·kg^−1^, respectively) was collected from the Agronomic Field Research Area stationed at the north campus of Northwest A&F University, China. A 2 mm sample of sieved air-dried soil, collected from various spots, was used for pot filling. The spiking of the potted soil was done using cadmium sulfate at 30 mg kg^−1^ soil. Melatonin (MEL), bought from a local market, was applied starting about 16 days prior to transplanting, with a concentration of 100 µmol, applied four times at four-days intervals. The treatments designated for this work were as follows: control (CK), sole Cd, sole Mel, and Cd + Mel. For the Ck group, only irrigation was supplemented, without Cd and Mel. These treatments, with three replicates each, were arranged according to a completely randomized design. The experimental pots with transplanted tobacco seedlings were kept in a greenhouse at Northwest A&F University, under a light intensity of 500–530 μmol m^−1^ s^−1^ and 70–72% relative humidity. Approximately 70% of the field capacity was maintained throughout the growth phase. These tobacco seedlings were allowed to grow for about 60 days after transplanting, after which they were harvested to obtain the required parameters.

### 4.2. Determination of Growth Indices

Harvested plants from each treatment were separated into roots, shoots and leaves. Immediately, their fresh weights were taken considering 1/10,000 weighing balance. Later, the samples were initially dried with absorbent paper and subsequently oven-dried at 65 °C until a constant weight was achieved.

### 4.3. Photosynthetic Traits Measurements

The procedure quoted by Wang et al. [48] was followed to appraise photosynthesis parameters, considering fourth true leaves (three replications; from the uppermost expanded leaves) after 60 days of stress treatments. The state-of-the-art portable LI 6800 fluorimeter (LI-COR, Lincoln, NE, USA) system was operated to take the values of Pn, Ci, Tr and Gs. Measurements were taken in the morning between 10 and 11 a.m. on clear weather days, with the system maintained at a light intensity of 1000 μmol m^−2^ s^−1^. Moreover, a carbon dioxide concentration of 400 cm^3^ m^3^ was provided to the system. Chlorophyll fluorescence, expressed as the ratio of variable to maximum fluorescence (Fv/Fm), was measured using a fluorimeter (model, Hudson, TX, USA). Seedlings were subjected to a 30 min dark period prior to measuring the maximum quantum efficiency of PSII (Fv/Fm = (Fm − Fo)/Fm). For pigment determination, 0.5 g of leaves (from three replicates) were assimilated in a 95% ethanol solution. After centrifuged (3000× *g*) for 12 min, the filtrated solution was used to obtain the supernatant, which was further analyzed spectrophotometrically at 663 and 645 nm wavelengths.

### 4.4. Reactive Oxygen Species Quantification and Confocal Visualization

The method quoted by Pilz et al. [49] was followed to determine MDA content and ROS levels. First, MDA content, as an indicator of lipid peroxidation, was quantified from shoot tissues and analyzed using a microplate reader at 532 and 600 nm wavelengths [50]. Next, ROS levels, including H_2_O_2_ and O_2_^•−^ , were calculated using kit-based methods. For H_2_O_2_ and O_2_^•−^ , analyzing kits, namely, Cat No. BC3590 and Cat No. BC 1295, respectively, were used according to manufacturer’s instructions. Later, using a photometric microplate reader (model UV-2600), the absorbances were evaluated at 415 nm and 530 nm wavelengths, respectively. Moreover, the laser confocal microscopy technique was used to perform in vivo O_2_^•−^ visualization. For that, a fluorescent probe was used, as previously recommended [51]. Briefly, small slivers of shoot tissue were prepared and initially washed with distilled water. The clean segments were then submerged in dihydroethidium solution (10 μM DHE was dissolved in 10 mM Tris-HCl buffer, pH 7.4 at 37 °C darkness for 30 min) for 1 h at normal temperature in the dark. Afterward, PBS solution was used to wash the samples thrice before being placed on the microscope slides. Coverslips were then used for visualization under a confocal microscope. Excitation with PMT detection (518 nm) was set at 500–600 nm [51].

### 4.5. Determibation of Enzymatic and Non-Enzymatic Antioxidants

The procedure described by Foster et al. [52] was followed to ascertain antioxidative activities. Tobacco leaves were collected and used immediately after harvest. These samples were ground and emulsified in phosphate buffer (50 mM). After centrifugation at 10,000× *g* for 25 min, the obtained supernatant was collected for further analyses. For the SOD enzyme assay, firstly, a reaction solution with phosphate buffer (0.5 M), methionine, and tetrazolium blue solution (750 μM) was developed. After adding EDTA solution, riboflavin (20 μM) and deionized water, about 3 mL of the reaction solution was taken and mixed with the supernatant. The test tubes containing the working solution were kept under light conditions for 25 min. A control group with 20 μL buffer and 3 mL of SOD reaction solution was considered for comparison. Later, the absorbance of both groups was taken at a 560 nm wavelength, spectrophotometrically. Lastly, the SOD activity was considered, corresponding to the following equation:SOD activity = (Aa + Ab) × V/W × 30 × Aa

Here, Aa and Ab show the absorbance rates for the control and sample tubes, respectively. V, W and T are the extraction solution volume, sample mass and light time, respectively.

For POD activity, initially, supernatant from the enzymatic solution was added to 2.5 mL guaiacol solution. Later, H_2_O_2_ solution was added before placing the solution in a water bath for 20 min. Finally, distilled water was added to the working solution, and the absorbance was recorded spectrophotometrically at 470 nm. The values were obtained following the equation below, considering the absorbance value of the blank sample tube (Ao) along with the values for the control and sample solutions:POD activity = [(Aa + Ab)/(Ao − Aa)] × W × T × 100

Here, T is the reaction time.

For CAT determination, about 200 mL of phosphate buffer was used to prepare the reaction solution. Next, a working solution was prepared by adding H_2_O_2_ solution to the obtained supernatant. Later, keeping PBS solution as a control, the absorbance of the working and control groups was recorded at 240 nm wavelength, with absorbance values taken at 40 s intervals (△A). The below equation was used for CAT activity (U g^−1^ min^−1^):CAT = △A/(mass sample × 100)

For APX activity, the reaction solution was prepared according to the previously reported method of Nakano et al. [53]. Furthermore, the modified method quoted by Griffith et al. [54] was considered for assessing GST activity. Firstly, fresh tobacco leaves were extracted in phosphate buffer (PB) and EDTA solutions. Next, the extracted solution was centrifuged at 10,000× *g* for 20 min. Then, the reaction solution was obtained by combining the enzyme extract with PB solution, CDNB, and GSH. This solution was then used to record the absorbance at 340 nm. The method quoted by Ahanger et al. [55] was followed to determine AsA content. Accordingly, using the enzyme extract, the absorbance was measured at a 530 nm wavelength.

### 4.6. Measurements of Cd Concentration

Tobacco roots and shoots were collected and cleaned initially with pure water and then with 5 mM calcium dichloride. The cleaned samples were dried at 70 °C and then ingested in 2 mL HNO_3_ at high temperature. This process continued until the root and shoot tissues became transparent. Later, double-distilled water was used to dilute the solution to 10 mL. This diluted solution was used to determine Cd concentrations on plasma atomic emission spectroscopy.

### 4.7. Analysis of Scanning Electron Microscopy

Fresh tobacco leaves were taken from the control, Cd and combined Cd and melatonin treatments. After removing their veins, the leaves were soaked in glutaraldehyde solution along with phosphate buffer (PB). This process continued for about 5 h. Later, washing was performed three times using the solely PB solution. Then, these samples were homogenized in OsO_4_ and PB solutions for 2–3 h following three-fold washing again with solely PB. Later, dehydration was done with different concentrations of ethanol prior to double dehydration with alcohol. Lastly, these samples were observed under SEM after coating with gold–palladium in ion sputter mode, as quoted earlier by Shuvaeva et al. [56].

### 4.8. qRT-PCR Analysis

Leaves of tobacco plants subjected to control, sole Cd, melatonin, and their combined treatments were considered for the extraction of total RNA using the Total RNA Kit. Firstly, a NanoDrop spectrophotometer was used to assess the quantity, quality, and probity of RNA. Later, Rescript II RT SuperMix reverse transcriptase was considered to obtain the first strand of cDNA, which was produced from 1 μg of total RNA with 20 μL reaction volume. Next, 96-well blocks on a CFX96 Touch Real-Time PCR System were considered to perform quantitative real-time PCR. For this, 2 × SYBR Premix UrTaq II was used with a total reaction volume of 20 μL, as described previously by Piaxao et al. [57]. In total, three reactions were conducted for individual samples. The NtActin gene served as a housekeeping gene and the 2^−ΔΔCT^ approach was considered to evaluate the relative expressions of the target genes.

### 4.9. Statistical Analysis

One-way analysis of variance was performed to study the impacts of melatonin and Cd treatments on tobacco seedlings and attained values of the recorded traits. The means of three replicates plus standard errors are presented in this work. These means were compared using the least significant difference (LSD) test (*p*-value = 0.05) in SPSS software (version 22.0) [58]. Graphical presentation was performed in Origin software (Model 2021), whereas mantel analysis was performed in R-studio.

## 5. Conclusions

In this work, the remediation effects of soil-applied melatonin on Cd toxicity were assessed. Cd treatment alone had severe effects on the growth, physiological and biochemical indices of tobacco seedlings. However, soil-applied melatonin markedly improved seedling phenotype, enhancing growth, pigment formation and photosynthetic rates. Moreover, melatonin supplementation reduced oxidative damage by maintaining a balance between ROS production and antioxidant activity under Cd stress. The positive influence of melatonin on photosynthesis and enzymatic activities was further supported by the high expression of relevant genes under Cd stress. Our results indicate that soil-applied melatonin exerts numerous beneficial effects on tobacco performance under Cd stress. Overall, this study proposes exploring melatonin as a potential enhancer of phytoremediation, aiming to improve plant tolerance, pollutant uptake, and antioxidant defenses. Melatonin application could increase phytoremediation efficiency by strengthening plants’ resilience and pollutant accumulation capabilities. However, the remediation mechanisms of melatonin supplementation, particularly at the molecular level, warrant further investigation to fully elucidate its role in alleviating Cd toxicity in tobacco. Moreover, future studies should explore the dose–response relationship of melatonin under different soil Cd concentrations and investigate its long-term effects on soil health and microbial communities in tobacco cultivation systems.

## Figures and Tables

**Figure 1 plants-13-03049-f001:**
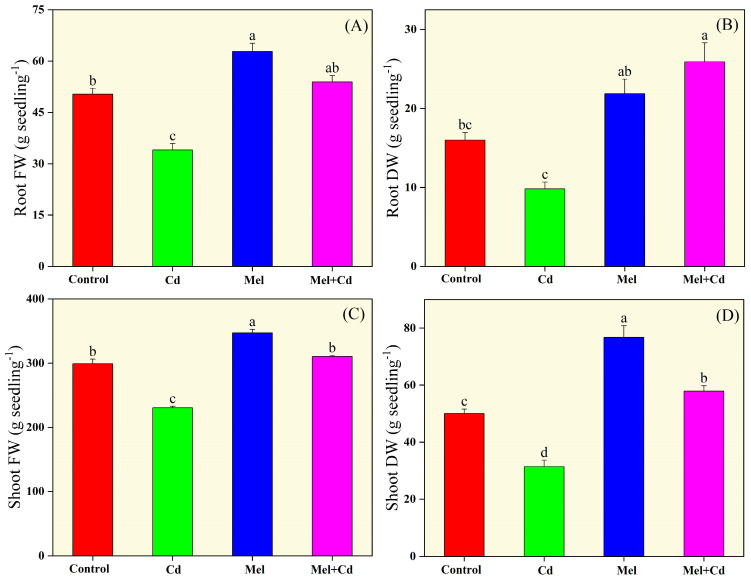
Effects of soil application of melatonin (Mel) on root fresh weight (FW; (**A**)), root dry weight (DW; (**B**)), shoot fresh weight (**C**) and shoot dry weight (**D**) of tobacco under cadmium (Cd) stress. The values of each parameter are means ± SE of three replicates. Similar letters show non-significant difference among the treatments according to LSD test (*p* < 0.05).

**Figure 2 plants-13-03049-f002:**
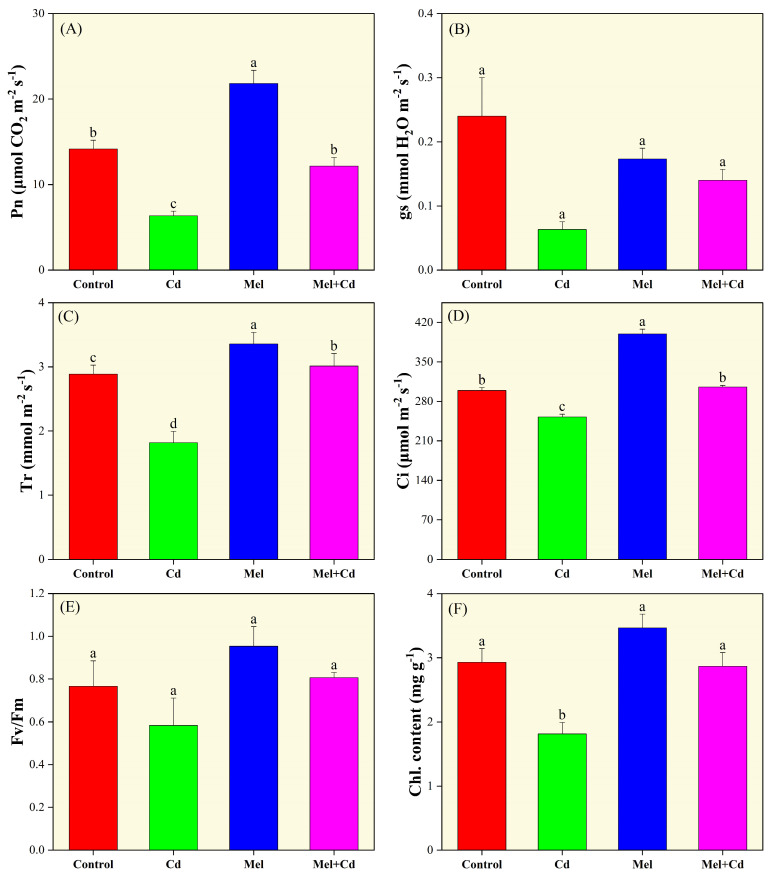
Effects of soil application of melatonin (Mel) on photosynthesis (Pn; (**A**)), stomatal conductance (gs; (**B**)), transpiration rate (Tr; (**C**)), intracellular CO_2_ concentration (Ci; (**D**)), chlorophyll florescence (Fv/Fm; (**E**)) and chlorophyll content (**F**) in tobacco seedlings under cadmium (Cd) stress. The values of each parameter are means ± SE of three replicates. Similar letters show non-significant difference among treatments according to LSD test (*p* < 0.05).

**Figure 3 plants-13-03049-f003:**
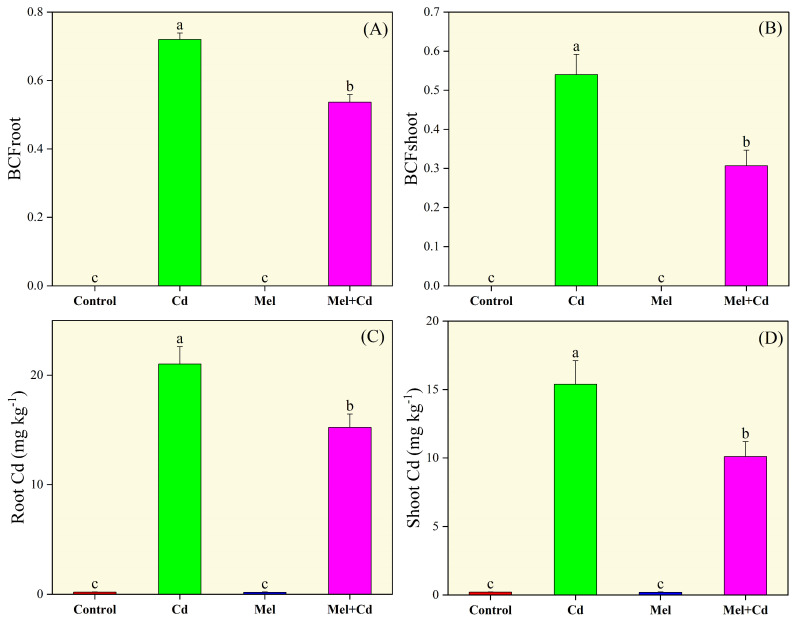
Effects of soil application of melatonin (Mel) on bioconcentration factor of root (BCFroot; (**A**)) and shoot (BCFshoot; (**B**)), cadmium (Cd) concentration in root (**C**) and shoot (**D**) of tobacco seedling under Cd stress. The values of each parameter are means ± SE of three replicates. Similar letters show non-significant differences among treatments according to LSD test (*p* < 0.05).

**Figure 4 plants-13-03049-f004:**
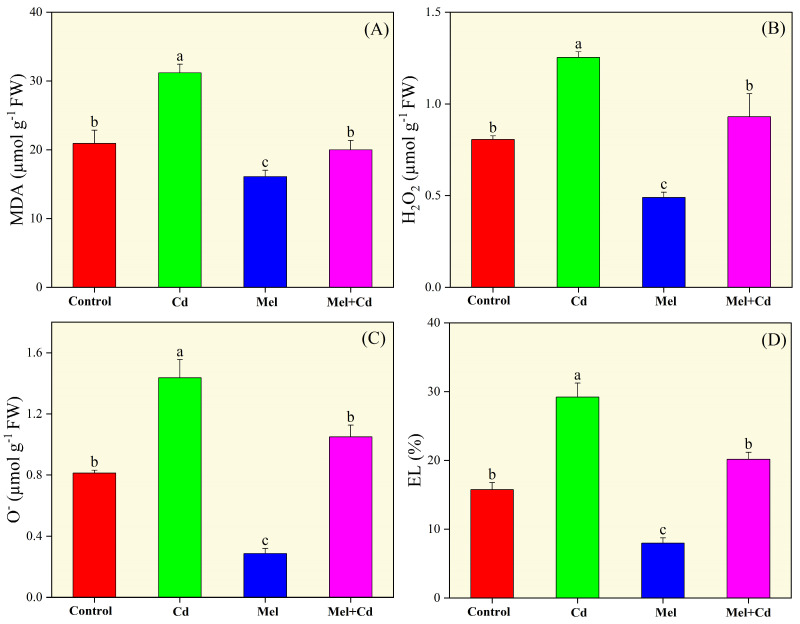
Effects of soil application of melatonin (Mel) on malondialdehyde (MDA; (**A**)), hydrogen peroxides (H_2_O_2_; (**B**)), superoxide anion radical (O_2_^•−^; (**C**)), and electrolyte leakage (EL; (**D**)) in tobacco seedling under cadmium (Cd) stress. The values of each parameter are means ± SE of three replicates. Similar letters show non-significant difference among treatments according to LSD test (*p* < 0.05).

**Figure 5 plants-13-03049-f005:**
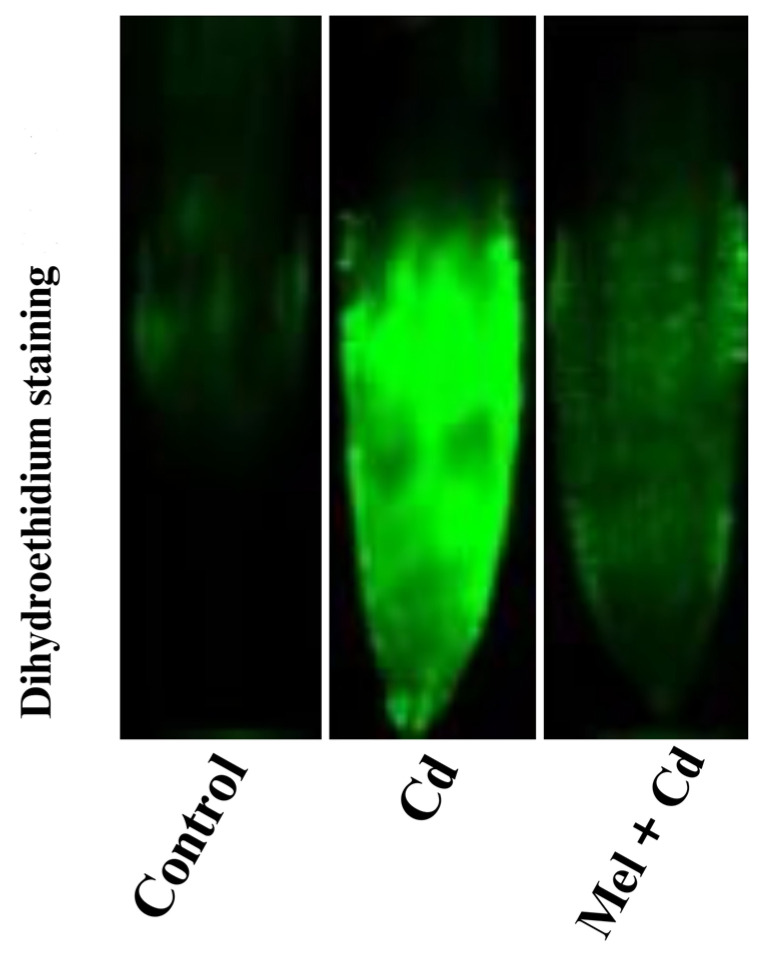
Tobacco leaves stained using dihydroethidium dye for O_2_^•−^ under control, cadmium (Cd), and melatonin (Mel) + Cd conditions.

**Figure 6 plants-13-03049-f006:**
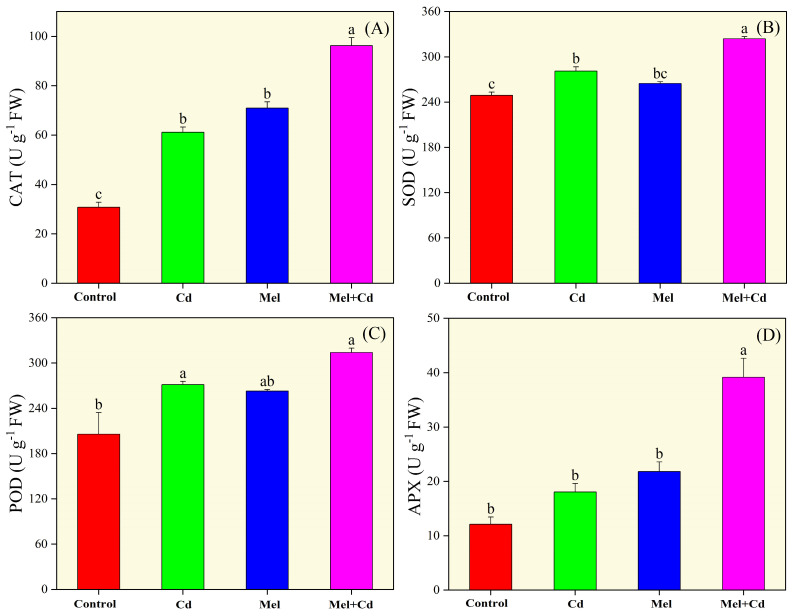
Effects of soil application of melatonin (Mel) on catalase (CAT; (**A**)), superoxide dismutase (SOD; (**B**)), peroxidase (POD; (**C**)), and ascorbate peroxidase (APX; (**D**)) in tobacco seedling under cadmium (Cd) stress. The values of each parameter are means ± SE of three replicates. Similar letters show non-significant difference among treatments according to LSD test (*p* < 0.05).

**Figure 7 plants-13-03049-f007:**
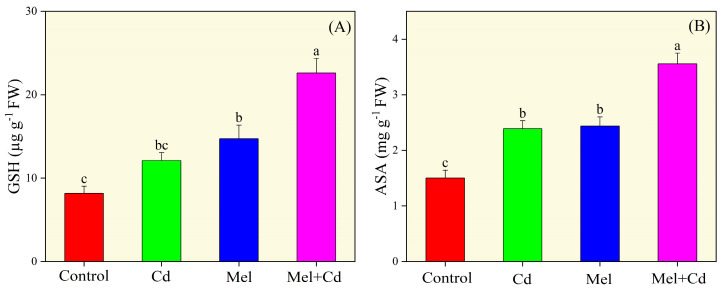
Effects of soil application of melatonin (Mel) on glutathione (GSH; (**A**)) and ascorbic acid (AsA; (**B**)) content in tobacco seedlings under cadmium (Cd) stress. The values of each parameter are means ± SE of three replicates. Similar letters show non-significant differences among treatments according to LSD test (*p* < 0.05).

**Figure 8 plants-13-03049-f008:**
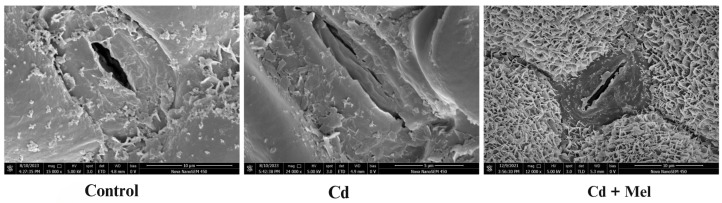
Effect of soil applied melatonin (Mel) on stomatal aperture and guard cells of tobacco leaves under cadmium (Cd) stress.

**Figure 9 plants-13-03049-f009:**
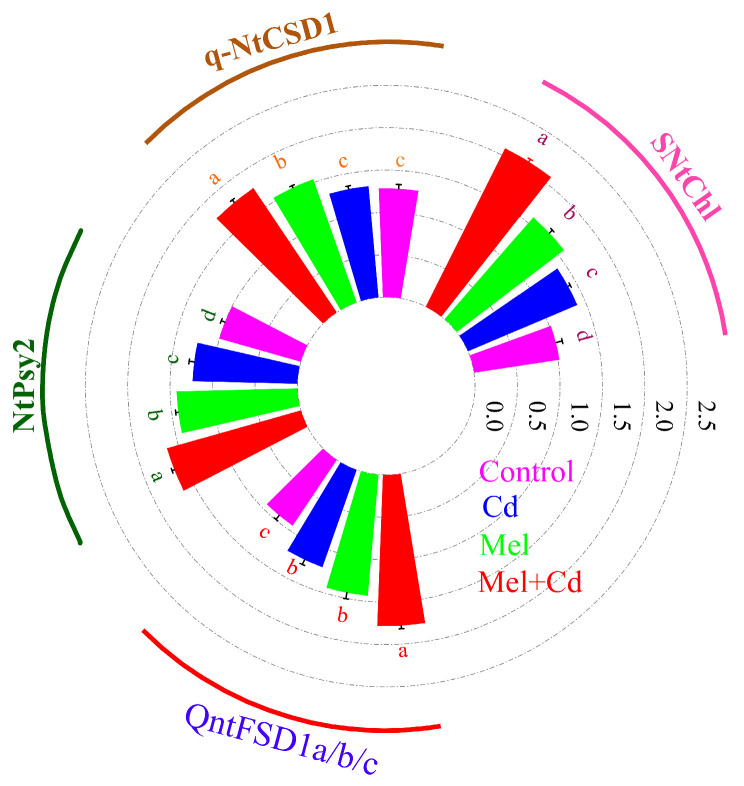
Effect of soil applied melatonin (Mel) on the expressions of various genes in tobacco leaves under cadmium (Cd) stress.

**Figure 10 plants-13-03049-f010:**
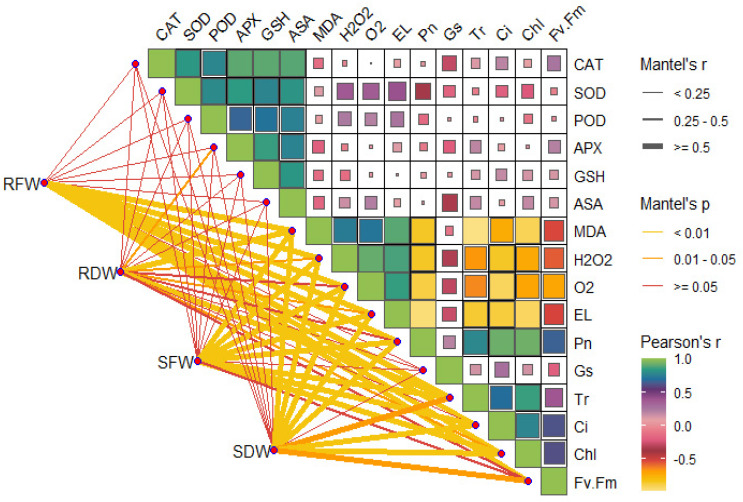
Mantel correlation coefficient among different growth, and physio-biochemical traits of tobacco under cadmium (Cd) stress and soil application of melatonin.

## Data Availability

All used data are given in the manuscript in the form of tables or figures.

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
