# Peer review of "Melatonin Ameliorates Cadmium Toxicity in Tobacco Seedlings by Depriving Its Bioaccumulation, Enhancing Photosynthetic Activity and Antioxidant Gene Expression"

_plants, 2024, doi:10.3390/plants13213049_

Round 1
Reviewer 1 Report
Comments and Suggestions for Authors
The manuscript presents a comprehensive study on the role of melatonin in mitigating cadmium (Cd) toxicity in tobacco seedlings. The authors have investigated the effects of soil-applied melatonin on Cd accumulation, growth parameters, photosynthesis, antioxidant enzyme activities, and gene expression in tobacco seedlings under Cd stress. Although many studies have reported on melatonin's role in mitigating cadmium (Cd) toxicity in plants, the findings of this study may contribute to the development of strategies for the remediation of Cd-contaminated soils and the improvement of crop quality and safety. My comments are as follows.
Line 143, the full name of BCF should be provided upon its first appearance, and the method section does not include any details regarding the calculation of BCF.
Change all instances of "CK" and "Ck" to "Ctrl" or "Control" throughout the manuscript, including in the figures.
Line 164, the full name of DHE should be provided upon its first appearance, and the method section should describe how the dihydroethidium solution is prepared and what the working concentration is.
Lines 165 and 179, although the text refers to "leaves," Figure 5 is labeled as "shoot," and the stained tissue in the figure does not appear to be leaves. Please clarify what tissue was used in the experiment.
SEM analysis. Specific statistical data on stomatal aperture opening should be provided, such as the average stomatal aperture diameter of 30 stomata.
Line 213, (Met) should be corrected to (Mel).
Based on the order of gene expression and SEM in the results, the order of 4.6. qRT-PCR analysis and 4.7. Analysis of Scanning electron microscopy in the methods should be swapped to match the order in the results.
Lines 228-237, What is meant by "Mental analysis"? Line 239, what is the "Mentel correlation coefficient"? How were the results in Figure 10 obtained? The authors did not provide any methodological details regarding Mantel analysis.
Line 304, what is referred to by "(Fig. X)"? I believe "(Fig. X)" should be a reference, such as a recent review, Noor, et al. 2024, From stress to resilience: Unraveling the molecular mechanisms of cadmium toxicity, detoxification, and tolerance in plants.
Line 329, the full name of "P. orientalis" should be provided upon its first appearance, and both it and "Malus domestica" should be italicized.
Lines 334-336, photos of the potted plants should be provided so that readers can understand the size and developmental stage of the tobacco seedlings used in the experiment, especially the phenotype of the seedlings after various treatments and the control.
In the discussion section, I notice that the authors only mention the similarities between the results of this study and previous studies, without discussing any differences or new findings compared to previous research.
In the qRT-PCR analysis, why not measure genes related to Cd transport and detoxification to better understand the molecular mechanisms underlying the ameliorative effects of melatonin?
In the evaluation of gene expression, the authors did not provide sufficient details regarding the specific genes studied (e.g., NtPsy2, QntFSD1), their functions, or why they were chosen. More context and rationale need to be given for the selection of these genes, along with a deeper discussion of how changes in their expression contribute to the observed phenotypes.
The authors should carefully proofread the manuscript for typographical errors and grammatical inconsistencies. The authors should ensure that terminology is used consistently and that abbreviations are defined at their first mention.
There are several typographical errors and inconsistencies throughout the text that need to be corrected.
Author Response
Line 143, the full name of BCF should be provided upon its first appearance, and the method section does not include any details regarding the calculation of BCF.
Response: Many thanks for highlighting this mistake. BCF indicates bioconcentration factor. The full name is now provided in the revised file.
Change all instances of "CK" and "Ck" to "Ctrl" or "Control" throughout the manuscript, including in the figures.
Response: Many thanks for your comments. Changes have been made in the revised file.
Line 164, the full name of DHE should be provided upon its first appearance, and the method section should describe how the dihydroethidium solution is prepared and what the working concentration is.
Response: Many thanks for your comments. The DHE full form has been provided in the revised file. Moreover, the detailed methodology has been added for better clarity.
“The clean segments were then submerged in dihydroethidium solution (10 μM DHE dissolved in 10 mM Tris-HCl buffer, pH 7.4 at 37℃ darkness for 30 min) for 1 hour at normal temperature in the dark. Afterward, PBS solution was used to wash the samples thrice before being placed on the microscope slides.”
Lines 165 and 179, although the text refers to "leaves," Figure 5 is labeled as "shoot," and the stained tissue in the figure does not appear to be leaves. Please clarify what tissue was used in the experiment.
Response: Many thanks for highlighting this mistake. Leaves were used for staining. The typo error in Figure 5 has been corrected in the revised file.
SEM analysis. Specific statistical data on stomatal aperture opening should be provided, such as the average stomatal aperture diameter of 30 stomata.
Response: Many thanks for your comment and suggestion. In this experiment, SEM analysis was performed only to visualize the stomatal opening or closing. We have not collected the statistical data. However, in future experiments, we should focus on the statistical data as well.
Line 213, (Met) should be corrected to (Mel).
Response: Thanks for highlighting this mistake. Done as suggested.
Based on the order of gene expression and SEM in the results, the order of 4.6. qRT-PCR analysis and 4.7. Analysis of Scanning electron microscopy in the methods should be swapped to match the order in the results.
Response: Thanks. Done as suggested.
Lines 228-237, What is meant by "Mental analysis"? Line 239, what is the "Mentel correlation coefficient"? How were the results in Figure 10 obtained? The authors did not provide any methodological details regarding Mantel analysis.
Response: Many thanks for your comments. We are sorry for the typo error, which is omitted in the revised file. Mantel analysis was used to clarify the relationship between growth indices and physio-biochemical traits. Basically, this analysis is used to assess the correlation between the recorded matrices. Replicated data was used to generate the Mantel test on R-studio (Figure 10). The same test was used in different recent studies (https://doi.org/10.1016/j.apsoil.2024.105639; https://doi.org/10.1016/j.apsoil.2024.105685).
Line 304, what is referred to by "(Fig. X)"? I believe "(Fig. X)" should be a reference, such as a recent review, Noor, et al. 2024, From stress to resilience: Unraveling the molecular mechanisms of cadmium toxicity, detoxification, and tolerance in plants.
Response: Many thanks for your comment and highlighting this mistake. Here, “Fig. X” was mentioned mistakenly. In the revised file we have changed “Fig. X” to “Fig. 6”.
Line 329, the full name of "P. orientalis" should be provided upon its first appearance, and both it and "Malus domestica" should be italicized.
Response: Many thanks for highlighting this mistake. Corrections have been made in the revised file.
Lines 334-336, photos of the potted plants should be provided so that readers can understand the size and developmental stage of the tobacco seedlings used in the experiment, especially the phenotype of the seedlings after various treatments and the control.
Response: Many thanks for your comments and suggestions. A picture of plants with pots has been provided in the revised file (please see supplementary file data).
In the discussion section, I notice that the authors only mention the similarities between the results of this study and previous studies, without discussing any differences or new findings compared to previous research.
Response: Many thanks for your comments. As per your suggestion, significant changes have been made in the revised file.
In the qRT-PCR analysis, why not measure genes related to Cd transport and detoxification to better understand the molecular mechanisms underlying the ameliorative effects of melatonin?
In the evaluation of gene expression, the authors did not provide sufficient details regarding the specific genes studied (e.g., NtPsy2, QntFSD1), their functions, or why they were chosen. More context and rationale need to be given for the selection of these genes, along with a deeper discussion of how changes in their expression contribute to the observed phenotypes.
Response: Thank you for your comments. In this study, our main focus was on examining the physiological and biochemical performance of tobacco seedlings under Cd stress. Therefore, based on the availability of funds, we selected genes related to photosynthesis (NtPsy2 and NtChl) and antioxidant enzyme activities (NtFSD1a/b/c and NtCSD1). However, in future studies, we will focus on additional genes as well.
The authors should carefully proofread the manuscript for typographical errors and grammatical inconsistencies. The authors should ensure that terminology is used consistently and that abbreviations are defined at their first mention. There are several typographical errors and inconsistencies throughout the text that need to be corrected.
Response: Many thanks for your suggestions. A thorough revision was made regarding the English language by a native English speaker, who is expert in this study area as well. The whole manuscript is revised now, and all changes are now omitted.
Reviewer 2 Report
Comments and Suggestions for Authors
Please read the attached file.

Author Response
Title of the work. You can make the title of the article more generalizing your work. For example - Melatonin regulates photosynthetic activity, antioxidant status and phytoremediation potential of tobacco under cadmium
Response: Thank you for your suggestions and comments. We have revised the title to “Melatonin ameliorates cadmium toxicity in tobacco seedlings by reducing its bioaccumulation, enhancing photosynthetic activity, and increasing antioxidant gene expression.” In the revised title, we changed “melatonin application” to “melatonin,” while the other terms remain unchanged, as the title was modified according to the editor's suggestion.
* Lines 44-45 check - sludge waste
Response. Thanks for your comment. Checked and satisfied.
*Please specify Keywords. Include -melatonin, antioxidant status.
Response: Thanks for your suggestion. Done as suggested.
* In the Introduction section. Clarify tasks 2) and 3). Can we say that genetic control of melatonin is part of the defense reactions? Lines 101-105
Response: Thank you for your suggestion. In this study, we specifically assess the expression of genes related to photosynthesis and antioxidants. To our knowledge, Objective 3 clearly conveys our intent: “to determine the relative expression of genes related to photosynthesis and enzymatic activities in tobacco exposed to Cd stress and melatonin treatment.”
*In the Materials and methods - Melatonin was used. If melatonin is added to the soil, what concentration did you use? How did you know that this concentration was suitable? Please check the cadmium concentration.
Response: Thank you for your comment. The selection of these (treatment) levels was based on a preliminary experiment in which various doses of melatonin viz. 0, 50, 100, 150, and 200 µmol were evaluated. According to the recorded data, superior seedling performance was noted when melatonin was applied at a dosage of 100 µmol, which was considered in this study. Moreover, higher concentrations (i.e., 150 and 200) caused toxic effects on tobacco seedlings (data not published).
“Melatonin (MEL), bought from a local market, was applied starting about 16 days prior to transplanting, with a concentration of 100 µmol, applied four times at four-days in-tervals”
* The text of the manuscript does not contain characteristics of the gene products that are studied in the work. This limits the understanding of the work. *Figure 9, there is (A), and where is (B)?
Response: Many thanks for your comments and suggestions. Genes products and their specific functions have been discussed in the revised file (please see lines 287-293 & 320-324). Thank you for highlighting the mistake in Figure 9, which is now revised for clarity.
*Melatonin is known to be an antioxidant. Discuss it here. How effective is the presence of melatonin in the soil in increasing the antioxidant status of plants, without stress. This is the key mechanism in this case. Here it is interesting to consider the ability of melatonin to regulate lignin deposition, and if there is any data in the literature, discuss it.
Response: Many thanks for your suggestions. In the revised file, the antioxidant properties of the melatonin have been discussed. Moreover, its ability to influence lignin deposition has also been elaborated.
*In the Conclusion or Abstract section, make a proposal on the possibility of using melatonin to enhance the phytoremediation properties of plants. This is very important. In this manuscript, tobacco cleans the soil from cadmium and grows well. This confirms the improvement of tobacco remediation properties with melatonin.
Response: Many thanks for your comments. The abstract and conclusion sections are now revised.
Round 2
Reviewer 1 Report
Comments and Suggestions for Authors
All my questions and concerns have been addressed.
Reviewer 2 Report
Comments and Suggestions for Authors
The authors of the manuscript did a good job. They significantly improved the materials. In the new version of the article I found other improvements that were apparently recommended by other reviewers. The manuscript is understandable. Thank you.